# Imaging Conductivity Changes in Monolayer Graphene Using Electrical Impedance Tomography

**DOI:** 10.3390/mi11121074

**Published:** 2020-12-01

**Authors:** Anil Kumar Khambampati, Sheik Abdur Rahman, Sunam Kumar Sharma, Woo Young Kim, Kyung Youn Kim

**Affiliations:** Department of Electronic Engineering, Jeju National University, Jeju 63243, Korea; anil@jejunu.ac.kr (A.K.K.); abdurrahman@jejunu.ac.kr (S.A.R.); sunamsharma@jejunu.ac.kr (S.K.S.); semigumi@jejunu.ac.kr (W.Y.K.)

**Keywords:** graphene, electrical conductivity, electrical impedance tomography, difference imaging, inverse problem

## Abstract

Recently, graphene has gained a lot of attention in the electronic industry due to its unique properties and has paved the way for realizing novel devices in the field of electronics. For the development of new device applications, it is necessary to grow large wafer-sized monolayer graphene samples. Among the methods to synthesize large graphene films, chemical vapor deposition (CVD) is one of the promising and common techniques. However, during the growth and transfer of the CVD graphene monolayer, defects such as wrinkles, cracks, and holes appear on the graphene surface. These defects can influence the electrical properties and it is of interest to know the quality of graphene samples non-destructively. Electrical impedance tomography (EIT) can be applied as an alternate method to determine conductivity distribution non-destructively. The EIT inverse problem of reconstructing conductivity is highly non-linear and is heavily dependent on measurement accuracy and modeling errors related to an accurate knowledge of electrode location, contact resistances, the exact outer boundary of the graphene wafer, etc. In practical situations, it is difficult to eliminate these modeling errors as complete knowledge of the electrode contact impedance and outer domain boundary is not fully available, and this leads to an undesirable solution. In this paper, a difference imaging approach is proposed to estimate the conductivity change of graphene with respect to the reference distribution from the data sets collected before and after the change. The estimated conductivity change can be used to locate the defects on the graphene surface caused due to the CVD transfer process or environment interaction. Numerical and experimental results with graphene sample of size 2.5 × 2.5 cm are performed to determine the change in conductivity distribution and the results show that the proposed difference imaging approach handles the modeling errors and estimates the conductivity distribution with good accuracy.

## 1. Introduction

Graphene, which is an allotrope of carbon, is a two-dimensional (2D) material made of a single atomic layer with carbon atoms arranged in a hexagonal honeycomb lattice and has special unique properties [1]. Graphene has remarkable electronic [2], mechanical [3], physical, and chemical properties [4]. For instance, at room temperature the carrier mobility is very high (>250,000 cm^2^V^−1^S^−1^), thermal conductivity (3000–5000 Wm^−1^K^−1^), high modulus (~1 TPa) despite small thickness (~3.4 Å), and high transparency nature [2,3,4,5]. The above properties make graphene ideal material for novel future device applications involving nanoelectronics, thin-film transistors, and transparent conductive electrodes for flexible and printable optoelectronics, and photovoltaic devices [6,7,8] to name a few. The significance of graphene for the development of these applications is the high electrical conductivity of the material. 

Graphene properties are dependent on the number and thickness of the graphene layers [9]. Moreover, it is difficult to grow graphene 2D crystals beyond small sizes as with increasing the lateral size the 2D crystallites bend into the third dimension. It is observed that interactions with 3D structures can stabilize the growth of 2D crystals therefore graphene can be grown or placed between two substrates or placed over the top of the bulk 3D structure [10]. Therefore, to develop graphene-based devices, it is necessary to grow large-area single-layer graphene on a suitable substrate and be able to control the process to achieve continuous production. Several methods are developed for the fabrication of graphene and are mainly classified as exfoliation, epitaxial growth, colloidal suspension, and unconventional methods [11]. Mechanical, thermal, or chemical energy is used to separate the stacked graphene layers from the bulk graphite in exfoliation [1,11,12,13]. In the epitaxial synthesis method, graphene is grown on the metallic or insulator substrate using physical or chemical vapor deposition. The colloidal suspension of graphene sheets is produced with initial raw material such as pristine graphite sheets or graphene oxide mixed with an aqueous or organic solvent. 

Among the developed methods, micromechanical cleavage produces high-quality graphene, but it can yield very small flakes of order few microns. For the device applications, large grown wafer-sized graphene samples are necessary. Among the different methods to obtain large graphene films, chemical vapor deposition (CVD) is a particularly promising technique [14,15]. However, during the growth and transfer process of CVD graphene films produces inhomogeneities such as cracks [16,17], wrinkles [18,19], and domain boundaries and affects the sheet resistance that can influence the electrical transport properties [16,18]. Therefore there is a need to map the local conductivity profile and characterize graphene film at a sub-micrometer resolution to associate graphene film and the various kinds of local defects for electrical characterization. 

In terms of electrical characterization for graphene, there is not much advancement in manufacturing methods to develop good throughput and quality. The popular way to evaluate the suitability of graphene films for a specific application is the four-point probe method or Hall effect measurements [16,18,20]. This is generally slow and inefficient and results in the destruction of the graphene film. Also, microscopic techniques such as Kelvin probe force microscopy [19,20,21], and conductive atomic force microscopy (AFM) [22], scanning tunneling microscopy (STM) [23,24] are used for studying the electrical properties of graphene locally at the nanoscale spatial resolution. Although they provide accurate characterization locally, to have the mapping of conductivity over a large area of graphene it is cumbersome and difficult to evaluate the sample quality in a given time frame. As a result, fast accurate method for a large area, high-density electrical property mapping is required for quality control, process optimization.

Non-contact methods for characterizing the electrical properties of graphene sheets covering a large area such as Terahertz domain spectroscopy (TDS) mapping [25,26], Infrared (IR) thermography [27,28], Lock-in thermography [29] are studied. However, it is sometimes difficult to identify local defects on a graphene sheet due to the low spatial resolution of TDS which is under a hundred micrometers to under a millimeter. Moreover, TDS measurement is obtained in a current-off condition, which does not indicate the true electrical properties of devices. Infrared (IR) thermography images large areas of graphene sample by the heat radiated from the biased sample. IR imaging can identify the carrier distribution and the electrical failures across the sample but due to the heat stored in the substrate that supports the graphene sample, the thermal radiation spot gets widened thus making it hard to locate defects at micrometer resolution. Electrical impedance tomography (EIT) which is a nonintrusive method can be used as an alternate method to determine conductivity mapping across the graphene surface. EIT is applied to several other applications in the area of process tomography [30,31,32,33] and biomedical applications [34,35,36]. For more information about EIT applications, see [37,38]. Defects such as cracks, holes, and wrinkles have different electrical properties as compared to the background region. This difference in electrical properties in the graphene sample can be identified using EIT by injecting currents and measuring voltages on electrodes that contact with the boundary of the graphene sample on the substrate. EIT hardware contains electronic components mainly fast multiplexers and analog to digital converters that have millisecond measurement time thus can be able to achieve real-time conductivity imaging of graphene samples. 

Recently, electrical impedance tomography (EIT) is applied as an alternate nondestructive method to map the conductivity distribution of graphene assuming point electrodes [39]. Point electrodes use the average gap model as a mathematical model which is not that accurate in real situations. For large size wafers point electrodes are not feasible and electrodes with width can be preferred. Moreover, the image is reconstructed using the Gauss–Newton method which is an absolute imaging method. In absolute imaging (ABI), the exact information regarding model parameters such as electrode location, contact impedance, and domain boundary is necessary to have a desirable solution with the absolute imaging approach [40,41]. However, in practical conditions, the knowledge about model parameters is uncertain and not complete. For example, the graphene sample that is transferred to the substrate has an outer boundary that is not uniform. The modeling errors and measurement noise can lead to severe errors in absolute reconstructions [42,43]. Difference imaging estimates the change in conductivity from the initial or reference distribution from the data that is measured before and after the conductivity change. Difference imaging uses data collected from the same geometry. Therefore, the modeling errors are compensated to some extent [44,45]. In this study difference imaging approach is applied to reconstruct the conductivity changes on the monolayer graphene sample using EIT. The forward problem is solved using the finite element method using boundary conditions applying a complete electrode model. The one step Gauss–Newton algorithm, which is a one-step algorithm, is used to compute the change in conductivity distribution. The change in conductivity can be used to locate the defects if present and provide information about the doping concentration which is useful in electrical characterization. Numerical and experimental studies are performed using the proposed method on large-area graphene of size 2.5 cm × 2.5 cm for conductivity estimation and the results are compared with the absolute imaging approach. 

## 2. Electrical Impedance Tomography

### Mathematical Model of Electrical Impedance Tomography (EIT)

In EIT to obtain the conductivity distribution of the graphene sheet, an array of *L* electrodes $e_{l}\;(l=1,2,\cdots ,L)$ is placed on the boundary of the graphene sample ∂Ω. An alternating current of small magnitude $I_{l}$ is injected through the electrodes on the graphene sample and the resultant voltages generated are measured across the electrodes. Using the measured voltages on the graphene and the associated injected currents, the conductivity distribution across the surface is determined (Figure 1). Reconstruction of conductivity using EIT is obtained by solving the forward and inverse problem iteratively until the desired solution is achieved. The forward problem of EIT here is to calculate the measured voltages from injected currents and given conductivity distribution σ(x,y) of the graphene sheet by solving the governing equation derived from Maxwell’s equation given by
(1)∇⋅(σ∇u)=0, in Ω
and boundary conditions that are defined based on the complete electrode model (CEM) given as [46]
(2)$$u+z_{l}\sigma \frac{\partial u}{\partial n}=V_{l},\;\left(x,y\right)\in e_{l},\;l=1,2,\cdots ,L,$$
(3)$$\int_{e_{l}}\sigma \frac{\partial u}{\partial n}dS=I_{l},\;\left(x,y\right)\in e_{l},\;l=1,2,\cdots ,L$$
(4)$$\sigma \frac{\partial u}{\partial n}=0,\;\left(x,y\right)\in \partial \Omega \backslash \overset{L}{\underset{l=1}{\cup }}e_{l}$$
where $z_{l}$ is the contact resistance of *l*th electrode, $V_{l}$ is the boundary voltage measured on *l*th electrode, n is the outward unit normal. Further, to have a unique solution, additional constraints are used for injected currents (Kirchoff current law) and measured voltages, i.e., fixing the reference potential level [47]
(5)$$\sum_{l=1}^{L}I_{{}^{l}}=0,\;\sum_{l=1}^{L}V=0$$

A numerical solution based on the finite element method is used to solve the forward problem. In the 2D finite element formulation, the computation domain i.e., graphene layer surface is discretized into triangular elements where within each element the resistivity distribution is assumed to constant. For the complete details about 2D finite element formulation for the EIT forward solution see [41]. Here it is discussed briefly again to have a better understanding of the inverse solution. Discretizing the computation domain Ω, the potential distribution is approximated as
(6)$$u\approx u^{h}\left(x,y,z\right)=\sum_{i=1}^{N}\alpha_{i}\varphi_{i}\left(x,y,z\right)$$

And the corresponding boundary voltages on electrodes are described as
(7)$$U^{h}=\sum_{j=1}^{L-1}\beta_{j}n_{j}$$
where $\alpha_{i}$, $\beta_{j}$ are the nodal potential and boundary voltages. Further, $\varphi_{i}$, *N* are basis functions and the number of finite element nodes considered, respectively. In Equation (7), $n_{j}$ are the measurement bases considered such that the constraint for voltage Equation (5) is satisfied. The boundary voltages $U_{L}$ on the electrode are calculated with the help of Equation (5) as
(8)$$\begin{array}{c} U_{1}=\sum_{l=1}^{L-1}\beta_{l}\\ U_{2}=-\beta_{1}\\ U_{3}=-\beta_{2}\\ \vdots \\ U_{L}=-\beta_{L-1} \end{array}$$

Using Equations (6) and (7) and the boundary conditions Equations (2)–(5), the FEM solution results in a set of algebraic equations and is represented in a matrix form
(9)$$Ab=\tilde{I}$$
where **A** is the system matrix, $b=\left(\begin{array}{c} \alpha \\ \beta \end{array}\right)$ is the unknown node potentials in the domain and boundary voltages on the electrode, $\tilde{I}=\left(\begin{array}{c} 0\\ \zeta \end{array}\right)$ is the data vector with $0\in {ℜ}^{N}$, $\zeta ={\left(I_{1}-I_{2},I_{1}-I_{3},\ldots ,I_{1}-I_{L}\right)}^{T}\in {ℜ}^{\left(L-1\right)}$. The FEM approximation of boundary voltages is represented as U(ρ) and the noise associated with the instrument during measurement (*e*) is considered to be white Gaussian, the EIT observation equation can be formulated as
(10)V=U(ρ)+e

To obtain the solution for the forward problem, Equation (9) is solved as $b=A^{-1}\overset{⌢}{I}$. 

## 3. Inverse Problem

For the conductivity estimation, let us consider the domain is discretized into small pixels as $\Omega =\cup_{n=1}^{N}\Omega_{n}$ and the conductivity distribution inside graphene sheet is assumed to be a pixel-wise constant such that
(11)$$\sigma (x)=\sum_{n=1}^{N}\sigma_{n}\chi_{\Omega_{n}}(x),$$
where the conductivity vector $\sigma ={\left[\sigma_{1},\sigma_{2},\cdots ,\sigma_{N}\right]}^{T}\in {ℜ}^{N\times 1}$ is the unknown to be estimated by minimizing the cost function that is the sum of squares of the voltage difference. 

### 3.1. Absolute Imaging

In EIT the inverse problem is ill-posed i.e., the number of voltage data (M) measured on the boundary is less than the number of unknowns (conductivity of each pixel *N*). Also, the Hessian matrix is ill-conditioned as the change in voltage due to conductivity in a given pixel is non-uniform. Hence, to have a stable solution, quite often regularization methods are used to reduce the ill-conditioned and ill-posedness of the EIT inverse problem. The regularized cost functional is therefore expressed as [40,41]
(12)$$\Phi (\sigma )=\frac{1}{2}{\left[U(\sigma )-V\right]}^{T}[U(\sigma )-V]+\frac{\alpha }{2}{\left[R(\sigma -\sigma^{*})\right]}^{T}[R(\sigma -\sigma^{*})],$$
where α is the smoothing or regularization parameter, $R\in {ℜ}^{N\times N}$ is the regularization matrix, and $\sigma^{*}\in {ℜ}^{N\times 1}$ is the available conductivity prior information. To solve the above nonlinear least-squares cost function, the Gauss–Newton method is often used. Starting with an initial distribution $\sigma_{0}$, the above cost function is minimized iteratively using the Gauss–Newton algorithm as follows
(13)$$\sigma_{{}^{k+1}}=\sigma_{{}^{k}}-{\left[J^{T}J+\alpha R^{T}R\right]}^{-1}\left[J^{T}(U(\sigma^{k})-V)+\alpha R^{T}R(\sigma -\sigma^{*})\right]$$
where J is the Jacobian defined as $J(\sigma_{{}^{k}})=\frac{\partial U(\sigma_{{}^{k}})}{\partial \sigma_{{}^{k}}}$. The conductivity reconstructed using Equation (13) can be termed as absolute imaging which gives the conductivity distribution at each pixel for the data set measured at a given time. 

### 3.2. Difference Imaging

In absolute imaging, the solution is highly dependent on the accuracy of forward solution and measured voltages. Apart from the measurement noise, there are uncertainties due to modeling errors associated with imprecise knowledge of electrodes position, the unknown shape of outer boundary, and contact impedances of electrodes [43]. The most often and commonly used method is the linear difference imaging (LDI) approach which is robust to modeling errors to an extent. Difference imaging estimates the conductivity change δσ based on the measurements $V_{1}$ and $V_{2}$ that relate to the data before and after the conductivity change at different time intervals
(14)$$V_{i}=U(\sigma_{i})+e_{i},\;i=1,2,$$
where $e_{i}$ is the Gaussian distributed measurement noise. Linearizing (14) and then subtracting, we have the expression for difference voltage data as
(15)$$V_{2}-V_{1}=\delta V=J\delta \sigma +\delta e,$$
where $\delta V=V_{2}-V_{1}$, $U(\sigma_{2})-U(\sigma_{1})≅J(\sigma_{2}-\sigma_{1})=J\delta \sigma$, and $\delta e=e_{2}-e_{1}$. The resistivity change from difference voltage data can be estimated by minimizing the cost function [44,45]
(16)$$\Phi (\sigma )=\arg\min\left\{{\left\|\delta V-J\delta \sigma \right\|}^{2}+\alpha {\left\|R\delta \sigma \right\|}^{2}\right\},$$
and assuming no prior knowledge about the conductivity, the solution for conductivity change is obtained as
(17)$$\delta \sigma ={\left(J^{T}J+\alpha R^{T}R\right)}^{-1}J^{T}\delta V$$

## 4. Results

This section presents the numerical and experimental studies to reconstruct the conductivity change across the graphene sample. The graphene sample used for the experimental and numerical case has the same geometry and the preparation of CVD graphene is described below

### 4.1. Simulation Studies

Numerical simulations are carried out using a graphene sample that is square-shaped and has the same geometry as the graphene sample used for the experiment. Fine mesh with 20,440 triangular elements is used to generate true data and coarse mesh with 5110 elements is used to estimate the conductivity distribution across the graphene (Figure 2). Here, two different meshes for forward and inverse computation are used so that inverse crime is avoided. Linear difference imaging is used as the inverse solution to map the conductivity profile and the results are compared with estimates using the absolute imaging method. Two numerical cases are considered for conductivity estimation where in the first case a homogeneous or ideal case is considered where the graphene has homogeneous distribution with conductivity 6.7 × 10^4^ mS/cm and data 2 contains a defect or crack that has conductivity 5 × 10^−9^ mS/cm on the surface. In the second test case, it is assumed the graphene sample has a defect that is like a wrinkle located at the bottom left side that has conductivity 2 × 10^3^ mS/cm, and after some time due to handling or transfer process, another defect is formed that has conductivity 5 × 10^−9^ mS/cm. The generated voltage data corresponding to data for the two cases is contaminated with 0.5% relative noise to account for instrument and numerical modeling errors.

#### Numerical Results

The results for conductivity estimation for numerical test case 1 are shown in Figure 3. The top row has the true distribution of conductivity for the initial homogeneous condition ($\sigma_{1}$) (Figure 3a) and data 2 has a defect located at the bottom left side (Figure 3b). The estimate of conductivities for $\sigma_{1}$ and $\sigma_{2}$ with conventional absolute imaging (ABI) is shown in Figure 3d,e. With ABI, it is noticed that estimates of $\sigma_{1}$ and $\sigma_{2}$ contain many artefacts. Especially, if we look at the estimated result $\sigma_{2}$, the low conductive region is seen at the bottom left side. However, estimates of ABI are affected due to measurement noise, and the size, location of the defect is not estimated with good accuracy. In this study, we are mainly interested to reconstruct the conductivity changes δσ. Solving the ABI estimates $\sigma_{1}$ and $\sigma_{2}$ separately and subtracting we can have the conductivity change, i.e., $\delta \sigma =\sigma_{2}-\sigma_{1}$ is computed and showed in Figure 3f. Conductivity change with ABI as similar to the estimated result $\sigma_{2}$ has an oversized low conductive region. Linear difference imaging (LDI) that estimates the conductivity change (δσ) using Equation (17) for case 1 is shown in Figure 3g. It is seen that the defect region with a low conductivity value is estimated with good accuracy. The location and size of the defect are estimated well with LDI however it is seen that the estimated shape of the defect is different from the true shape. This is due to the regularization method and the FEM approximation that is used in the forward solution to estimate the conductivity. To have a better estimation of defect shape, parametric methods that describe the shape of an object can be used, which is not within the scope of this work. 

Numerical results for conductivity estimation of case 2 with noise are shown in Figure 4. The true conductivity distribution is shown in the top row (Figure 4a–c). The initial state $\sigma_{1}$ has a small wrinkle defect located at the bottom left side of Figure 4a and the distribution $\sigma_{2}$ has two defects on the graphene surface Figure 4b. The conductivity estimates for $\sigma_{1}$, $\sigma_{2}$ and δσ with ABI and LDI is given in Figure 4d–g. It is seen that the small wrinkle defect is not identified with ABI in the estimates of $\sigma_{1}$ and $\sigma_{2}$
Figure 4d,e. Moreover, the background has many artefacts and does not give desirable information about the conductivity distribution. The conductivity change estimate δσ obtained by solving for $\sigma_{1}$ and $\sigma_{2}$ separately and then subtracting $\left(\sigma_{2}-\sigma_{1}\right)$ is shown in Figure 4f. It is found that the ABI estimate of conductivity change could locate the defect with a slightly bigger size. The estimate with the proposed LDI method is shown in Figure 4g and it could locate the defect and estimate the conductivity change with good accuracy. In the numerical simulation, it is assumed that contact resistances are assumed to be known however the measurement noise has a significant effect on ABI estimates, and with LDI the modeling errors are compensated to some extent. 

### 4.2. Experimental Studies

#### 4.2.1. Graphene Sample Preparation

For experiment purposes, the single-layer graphene synthesized on copper foil using the CVD process was purchased from Graphene Square Inc. As a mechanical supporter during the transfer process, poly(methyl-methacrylate) PMMA (950 PMMA A4, MicroChem Co., Westborough, MA, USA, was spin-coated on the graphene-synthesized Cu foil with the condition of spin speed of 3000 rpm for 30 s, and annealed on a hot plate at 100 °C for 5 min. Thereafter, PMMA/Graphene/Cu-foil was moved onto ammonium persulfate solution to dissolve the copper foil. After removing the whole copper foil, the PMMA/Graphene was cleaned using de-ionized water. Next, the PMMA/Graphene was transferred onto SiO_2_/Si wafer, target substrate. Finally, acetone was used to dissolve the PMMA. The graphene sample is now ready to use for experimental purposes. In this study, we have used a graphene sample of size 2.5 × 2.5 cm for examining the conductivity distribution using EIT. The graphene sample is then coated with copper electrodes using sputtering with a mask designed with the electrode location and shape. In Figure 5a, the graphene sample with copper electrodes used in the experiment is shown below. The graphene with copper electrodes is then connected to the EIT measurement setup with the help of gold wires as shown in Figure 5b. At both ends of gold wires, silver paste is used as an adhesive to connect copper electrodes on the graphene surface and copper wire terminals to connect to the EIT measurement setup.

To check the graphene sample, at four different locations as shown in Figure 6a Raman scanning is performed by exciting 532-nm laser. The spectra obtained at the four different data points are shown in Figure 6b–e. From the spectra, it can be seen that the G-band and 2D-band have a sharp band and the intensity ratio of the 2D-band to G-band is greater than 1. This can validate that the sample used is monolayer graphene. 

#### 4.2.2. EIT Measurement Setup

Laboratory experiments are performed using the EIT experiment setup that consists of Agilent 4284A precision LCR meter used as a constant current source and the national instrument system (NI PXI-1042Q National Instruments Corporation, Austin, TX, USA) is used as a data acquisition system to measure the resultant voltages. A total of 16 electrodes, four on each side of the graphene sample with electrodes placed equidistant on the boundary with a gap of 0.5 cm between them. As a current injection method, the adjacent method is used where neighboring electrodes are used for current injection and the resultant voltages across the other electrodes are measured. For a 16-electrode setup, with 0.1 mA injected current across the neighboring electrodes, there are 256 measurements for each data frame. 

#### 4.2.3. Experimental Results

EIT experiments are performed with a graphene sample of size 2.5 cm × 2.5 cm. Initial or reference data is obtained from a graphene plate without any external defects. After that on the graphene wafer, defects are made manually using a sharp pointed knife on the surface to investigate the conductivity distribution when the defects are present. In scenario 1 (Figure 7), we have the initial data $\left(\sigma_{1}\right)$ and after change data $\left(\sigma_{2}\right)$ with a single defect located near electrode 7 at the right side. The photograph of true experimental scenario is shown in Figure 7a,b. The ABI estimates for initial and with a single defect are shown in Figure 7c,d. The conductivity change estimate δσ for ABI which is shown in Figure 7e is obtained by subtracting the estimates $\left(\sigma_{2}\right)$ and $\left(\sigma_{1}\right)$. From Figure 7c,d, it is noticed that the ABI estimates for $\left(\sigma_{1}\right)$ and $\left(\sigma_{2}\right)$ has low conductivity distribution on the left side of the graphene sample and higher conductivity on the right side. The defect that was present with data $\left(\sigma_{2}\right)$ was not detected with ABI. The estimated conductivity change δσ has lower conductivity distribution on the right side of the graphene sample but it failed to identify the location of the defect (Figure 7e). In the experiment case, the contact impedance at each electrode is not the same and is unknown and the outer domain boundary of the graphene wafer is not exactly square. Due to this, the modeling errors in the experiment case are more as compared to the numerical case and have a significant effect on the estimates of ABI. The estimate of conductivity change with linear difference imaging (LDI) is shown in the third row and it is noticed that the low conductive region is estimated near electrode 7 which can be identified as a defect (Figure 7f). In LDI, modeling errors are compensated to some extent as the measurements obtained using the same domain are used in image reconstruction.

The experiment results for scenario 2 that has initial data and after change data with two defects are shown in Figure 8. The true experiment condition is displayed in the images in Figure 8a,b. ABI estimates could not identify the defects as noticed from the estimates shown in Figure 8c,d. Also, the estimate of conductivity change with ABI does not provide any information about the defects (Figure 8e). On the contrary, the estimates of LDI could identify and locate the two defects with good accuracy (Figure 8f).

The results for scenario 3 with initial and after change data containing three defects are shown in Figure 9. Figure 9a,b contains the true image of the experiment situation with initial data without defects and after change data with three defects on the graphene surface. In Figure 9c–e, the conductivity estimates of ABI for $\sigma_{1}$, $\sigma_{2}$, and δσ are shown. There is not much difference in the estimated conductivity for $\sigma_{1}$ and $\sigma_{2}$ using ABI, and defects are not identified. The LDI estimate (δσ) is shown in Figure 9f and it could identify all three defects. Thus, the estimated conductivity profile with ABI does not provide useful information about the conductivity behavior on the graphene surface if modeling errors are not compensated. LDI is seen to produce estimates that are close to true conditions and is tolerant of modeling errors to some extent. Therefore, in situations where initial or reference data is available LDI can be used to determine the conductivity changes that indicate the doping concentration on the graphene sheet. 

## 5. Conclusions

In this paper, we present electrical impedance tomography (EIT) based non-destructive electrical characterization method to determine the conductivity distribution of large-area monolayer graphene. Electrodes are placed on the boundary of the graphene outer boundary and by using the current-voltage relationship. The conductivity distribution can be reconstructed by solving the EIT forward and inverse solution. In practical situations, the exact knowledge about the electrode locations, contact impedances, and the outer boundary of the graphene is not known accurately. The associated modeling errors and measurement noise can affect the estimates of absolute imaging. Therefore, in this paper, a difference imaging-based approach is proposed to estimate the conductivity change based on the data corresponding initial and after the change. Difference imaging uses the data corresponding same geometry therefore the modeling errors are compensated to some extent and are useful to determine the conductivity profile of graphene. From the numerical and experimental studies with graphene wafer of size 2.5 × 2.5 cm it is found that LDI based conductivity estimates could identify the conductivity change and locate the defects with good accuracy as compared to ABI based solution that is affected due to modeling errors. 

LDI provides improved results and is computationally efficient as it is one-step and does not need any iterations however the solution is dependent on nominal or reference value and for situations where conductivity change is not very high. Absolute imaging of the conductivity profile is more practically useful for assessing the electrical properties and quality. However, the modeling errors have to be compensated. The simultaneous estimation of model parameters along with conductivity and use of the approximation error method can reduce the modeling errors in absolute imaging of the graphene conductivity profile, which will be studied in the future. 

## Figures and Tables

**Figure 1 micromachines-11-01074-f001:**
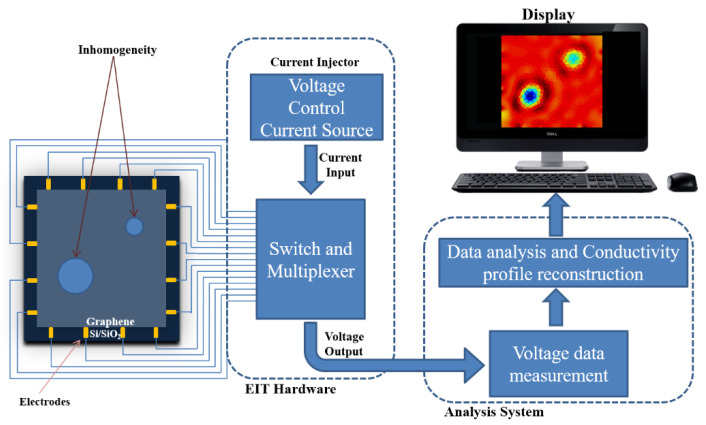
Diagram illustrating the application of electrical impedance tomography (EIT) to map the conductivity profile of monolayer graphene.

**Figure 2 micromachines-11-01074-f002:**
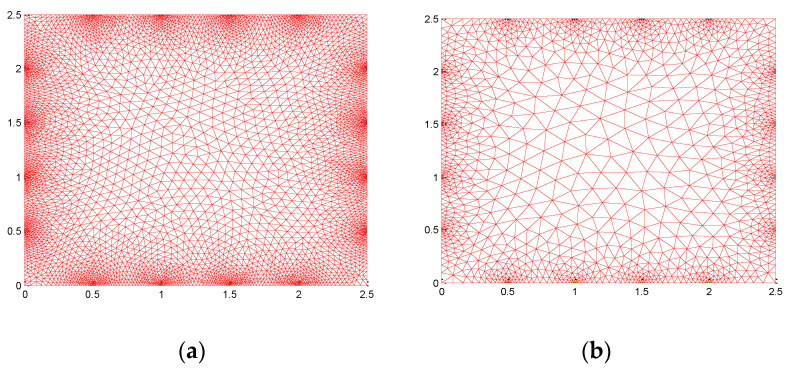
Finite element mesh used for numerical simulation (**a**) fine mesh for computing forward solution (**b**) coarse mesh used for the inverse problem.

**Figure 3 micromachines-11-01074-f003:**
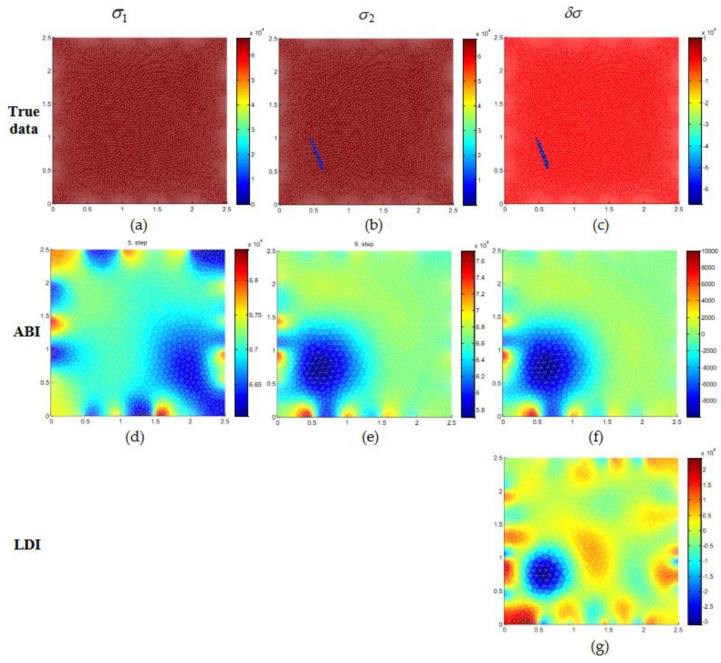
Numerical simulation of graphene conductivity estimation for test case 1 with 0.5% relative noise. (**a**–**c**) The top row is the true conductivity distribution while the second row (**d**–**f**), third row (**g**) are estimated conductivities of initial distribution $\left(\sigma_{1}\right)$, final distribution $\left(\sigma_{2}\right)$, and conductivity change (δσ) using absolute imaging (ABI) and linear difference imaging (LDI).

**Figure 4 micromachines-11-01074-f004:**
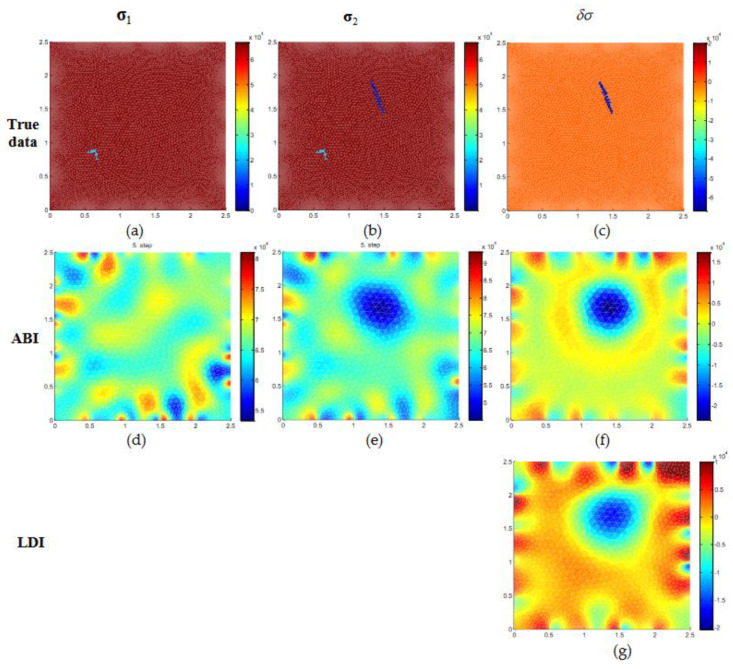
Numerical simulation of graphene conductivity estimation for test case 2 with 0.5% relative noise. (**a**–**c**) The top row is the true conductivity distribution while the second row (**d**–**f**), third row (**g**) are estimated conductivities of initial distribution $\left(\sigma_{1}\right)$, final distribution $\left(\sigma_{2}\right)$, and conductivity change (δσ) using ABI and LDI.

**Figure 5 micromachines-11-01074-f005:**
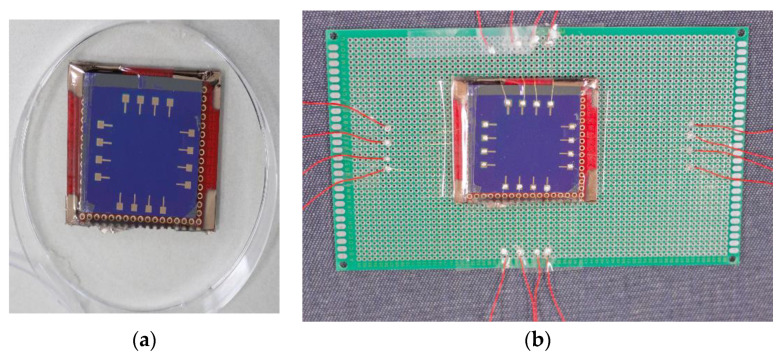
Graphene sample that is used in experimental studies for the conductivity profile. (**a**) copper electrodes coated on graphene surface (**b**) gold wires are used to connect the copper electrodes with copper wire terminals for EIT measurement setup.

**Figure 6 micromachines-11-01074-f006:**
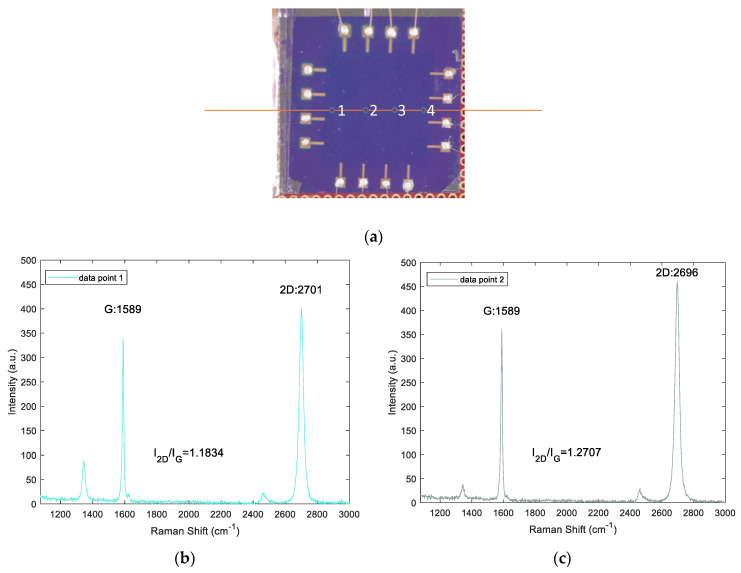
Raman spectra of monolayer graphene used for experimental studies (**a**) Locations of data points on graphene sample used for Raman scanning (**b**–**e**) Measured spectra for the locations of data points.

**Figure 7 micromachines-11-01074-f007:**
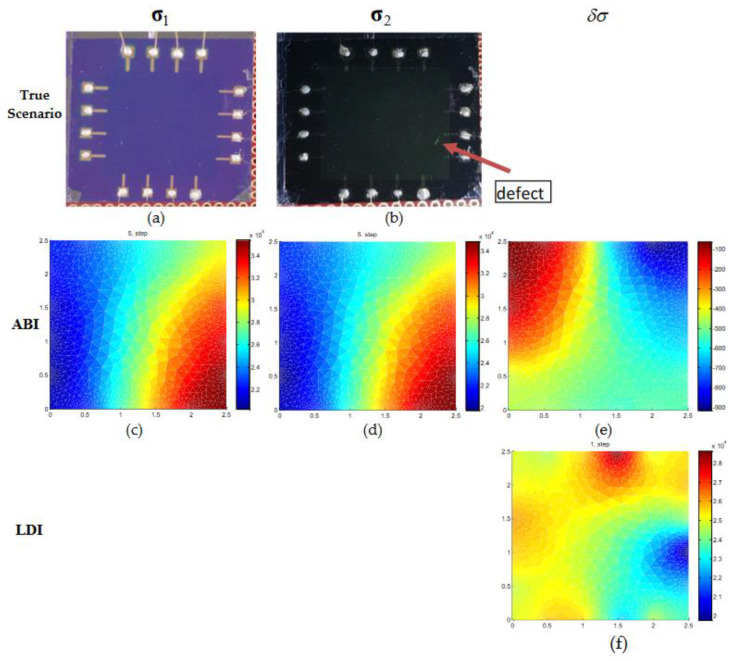
Conductivity estimation of real graphene wafer of size 2.5 × 2.5 cm. (**a**,**b**) The top row is the true condition of the real graphene sheet without defects and with a single defect used during the experiment. (**c**–**e**) The second row, (**f**) third row are the estimated conductivities of initial distribution $\left(\sigma_{1}\right)$, final distribution $\left(\sigma_{2}\right)$, and conductivity change (δσ) using ABI and LDI.

**Figure 8 micromachines-11-01074-f008:**
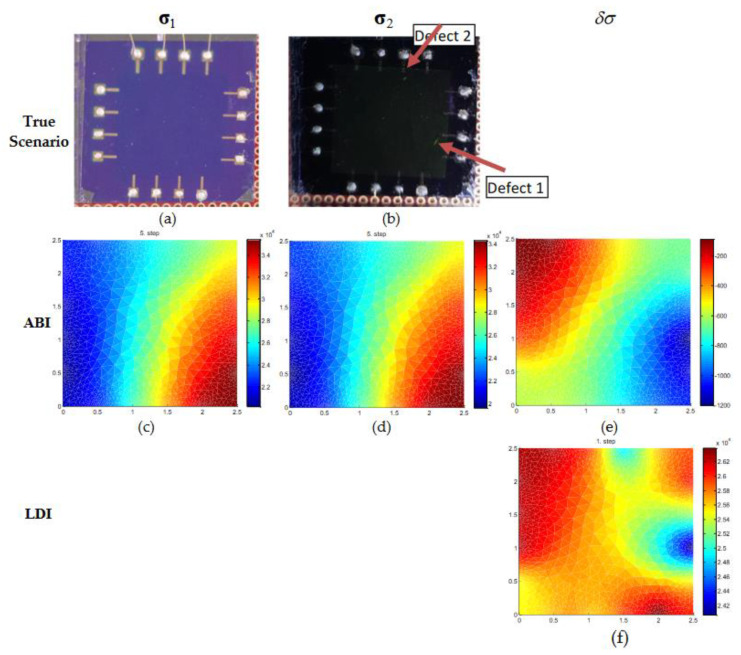
Conductivity estimation of real graphene wafer of size 2.5 × 2.5 cm. (**a**,**b**) The top row is the true condition of the real graphene sheet without defects and with two defects used during the experiment. (**c**–**e**) The second row, (**f**) third rows are the estimated conductivities of initial distribution $\left(\sigma_{1}\right)$, final distribution $\left(\sigma_{2}\right)$, and conductivity change (δσ) using ABI and LDI.

**Figure 9 micromachines-11-01074-f009:**
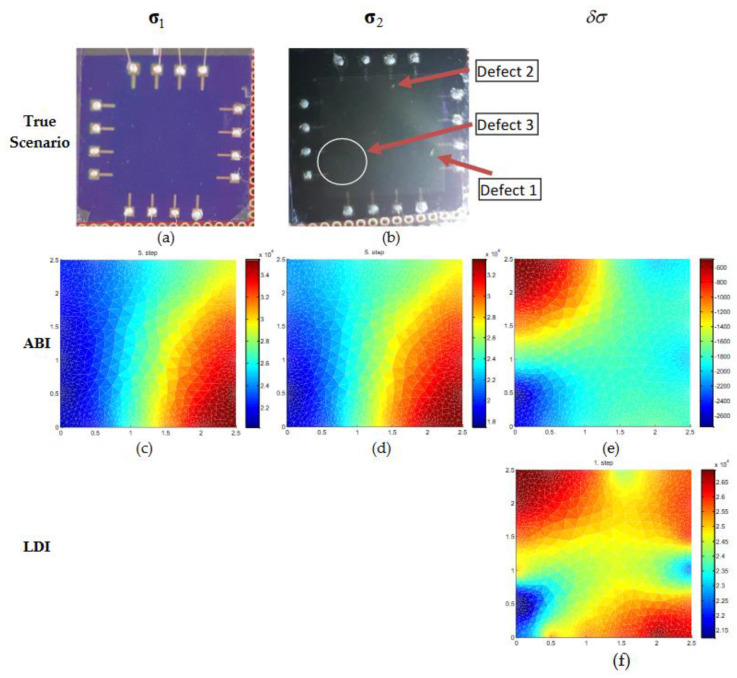
Conductivity estimation of real graphene wafer of size 2.5 × 2.5 cm. (**a**,**b**) the top row is the true condition of the real graphene sheet without defects and with three defects used during the experiment. (**c**–**e**) the second row, (**f**) the third row is the estimated conductivities of initial distribution $\left(\sigma_{1}\right)$, final distribution $\left(\sigma_{2}\right)$, and conductivity change (δσ) using ABI and LDI.

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
