# Peer review of "Imaging Conductivity Changes in Monolayer Graphene Using Electrical Impedance Tomography"

_micromachines, 2020, doi:10.3390/mi11121074_

Round 1
Reviewer 1 Report
This is avery interesting work, but the following details should be precised:
1) The authors should provide a further characterization of graphene sample (at least optical microscope/Raman spectra) to certify the quality of graphene. The PMMA residuals on graphene can be easily visualized with optical microscope and these iamges have to be provided to the reader.
2) The authors are speaking about defects near the electrode but they should to precise how this defect is created and how they know that there is a defect near the electrode. The authors should also provide an evidence of the defect existence with a different technique.
Author Response
The reviewers comments are upload as word file. Please see the attachment

Reviewer 2 Report
In this work by Khambampati et al., the authors propose to image conductivity changes in graphene. The work targets an important issue regarding its characterization. However, there are several major shortcomings, which must be handled before the work can be recommended for publication in Micromachines. Please find the suggestions below.
1) The level of English should be improved. There are many typos and grammar errors throughout the file. Please proofread the manuscript carefully to eliminate them.
2) The introduction section lacks references. Only 7 works support the first page of the manuscript. Please add necessary references to the literature to support many claims, which have been given.
3) Graphene preparation methodology should be a part of the Experimental section. Currently, it is given in the Results section.
4) There is insufficient information regarding the synthesis of graphene. The dimensions of the setup, concentrations, flow rates, etc., were not reported, making this study irreproducible.
5) The manuscript also lacks details regarding the material transfer process.
6) Formatting of the plots is not correct. The font size should be larger to make the data legible. What is more, the arrangement of the panels is not informative.
7) There is no information whatsoever regarding the characterization of graphene. Therefore, there is no certainty that indeed it was graphene. Please provide at least Raman spectra, SEM/TEM micrographs, TGA thermograms, XPS spectra, etc., to validate that it was, in fact, graphene. If so, it is of utmost importance to gather this data to know the quality of the material. The presence of defects and impurities can make an enormous impact on the properties. It is incorrect to claim that the material is "monolayer graphene" (taken from the title) without providing evidence.
Author Response
The comments to the reviewer are addressed and uploaded as a word file.
Please see the attachment

Round 2
Reviewer 2 Report
Thank you for following the recommendations. I suggest the article be accepted for publication in the journal.